# Impact of Microbiota: A Paradigm for Evolving Herd Immunity against Viral Diseases

**DOI:** 10.3390/v12101150

**Published:** 2020-10-10

**Authors:** Asha Shelly, Priya Gupta, Rahul Ahuja, Sudeepa Srichandan, Jairam Meena, Tanmay Majumdar

**Affiliations:** National Institute of Immunology, Aruna Asaf Ali Marg, New Delhi 110067, India; shelly.g341@gmail.com (A.S.); priya@nii.ac.in (P.G.); rahul74@nii.ac.in (R.A.); sudeepasrichandan@gmail.com (S.S.); jairam.meena20@gmail.com (J.M.)

**Keywords:** herd immunity, SARS-CoV-2, microbiome, nutrition, EE, heterologous immunity

## Abstract

Herd immunity is the most critical and essential prophylactic intervention that delivers protection against infectious diseases at both the individual and community level. This process of natural vaccination is immensely pertinent to the current context of a pandemic caused by severe acute respiratory syndrome coronavirus-2 (SARS-CoV-2) infection around the globe. The conventional idea of herd immunity is based on efficient transmission of pathogens and developing natural immunity within a population. This is entirely encouraging while fighting against any disease in pandemic circumstances. A spatial community is occupied by people having variable resistance capacity against a pathogen. Protection efficacy against once very common diseases like smallpox, poliovirus or measles has been possible only because of either natural vaccination through contagious infections or expanded immunization programs among communities. This has led to achieving herd immunity in some cohorts. The microbiome plays an essential role in developing the body’s immune cells for the emerging competent vaccination process, ensuring herd immunity. Frequency of interaction among microbiota, metabolic nutrients and individual immunity preserve the degree of vaccine effectiveness against several pathogens. Microbiome symbiosis regulates pathogen transmissibility and the success of vaccination among different age groups. Imbalance of nutrients perturbs microbiota and abrogates immunity. Thus, a particular population can become vulnerable to the infection. Intestinal dysbiosis leads to environmental enteropathy (EE). As a consequence, the generation of herd immunity can either be delayed or not start in a particular cohort. Moreover, disparities of the protective response of many vaccines in developing countries outside of developed countries are due to inconsistencies of healthy microbiota among the individuals. We suggested that pan-India poliovirus vaccination program, capable of inducing herd immunity among communities for the last 30 years, may also influence the inception of natural course of heterologous immunity against SARS-CoV-2 infection. Nonetheless, this anamnestic recall is somewhat counterintuitive, as antibody generation against original antigens of SARS-CoV-2 will be subdued due to original antigenic sin.

## 1. Introduction

Herd immunity, or community immunity, is fundamentally a concept of acquiring immunity through natural infection, or mass vaccination, in a particular population. While the purpose of individual vaccination is to prevent or reduce the chances of recurrent infection, in public health, the goal of herd immunity is to increase immunization efficacy to control or eradicate the infection in a particular cohort. Herd immunity is evidently a sequential process, because natural or vaccine-based immunity is lost over time through the waning of immune individuals due to death, and arrival of newly susceptible individuals due to birth or migration. Thus, to sustain herd immunity, it is important to vaccinate at regular intervals [1]. Herd immunity recuperates the protection against pathogens that are contagious in nature [2,3,4]. The successful eradication of several viral diseases such as smallpox [5] and poliovirus, reduction in transmission of pertussis and protection against influenza, pneumococcal disease, cholera [6] and rotavirus [7] was possible only through expanded immunization programs that have been fundamental factors for evolving herd immunity [8]. In addition, the level of vaccination needed to be achieved through herd immunity varies depending upon the frequencies of secondary infections [3,9]. For example, in measles, a highly contagious viral disease, one person can infect up to 18 individuals. Thus, 95% of the people are required to be immune in order to achieve herd immunity. The new SARS-CoV-2 has a lower infection rate than measles. On average, each infected person can pass the virus to two or three new people [10]. This means that herd immunity should be achieved when around 60% of a particular population becomes exposed to coronavirus disease 2019 (COVID-19) [11].

Here, we outline several factors that can directly or indirectly alter the acquisition of immunity. Among those factors, the gut microbiota is an essential element that contributes to shaping individual immunity and thereby mapping the population immunity. Gut microbiota, along with nutrients and environmental factors, play an indispensable and cumulative role in establishing herd immunity against any infectious disease. We propose that heterologous immunity is the most important factor contributing protection against SARS-CoV-2 infection in India. The robust poliovirus vaccination program developed herd immunity. This same method has the ability to deliver the protection against coronavirus infection.

## 2. Trinity of the Immune System Development: Microbiome, Nutrients and Environmental Factors

### 2.1. Putting the Microbiome, the Second Brain to Develop the Immune System

To generate efficient herd immunity, the maturation of the immune system is an essential and fundamental factor. Alterations of the microbiota by any means could pose an negative impact on the immune system’s development which upsets the vaccination program and subsequently, herd immunity [12,13]. Collectively, the microbial community residing in our body is termed as the microbiota. Microbiota, the ‘second brain’ of the body, outnumber the total cells of the human body by several fold. Ninety-nine percent of the total genome of human comes from microbiota. One percent comes from 23,000 genes of the body’s own cells [14,15,16]. The gut microbiota dysbiosis significantly affects the development and function of both innate and adaptive immunity [17,18]. The human microbiome regulates several functions of the body such as nutrient metabolism, intestinal barrier functions, shaping of the immune system, etc. It supports the prevention of several diseases and contributes to improving the genetic diversity in the population [19]. The absence of normal microbiota aggravates the maturation of the immune system, affecting both structure and functions as suggested by studies in germ-free (GF) mice [20,21]. However, by restoring microbiota symbiosis, all these effects can be reinstated.

In addition, the microbiome seems to provide a plausible explanation about the differential response of vaccination at both individual and population level. Gut microbiota have specific immunomodulatory properties [22]. *Bacteroides fragilis*, a Gram-negative bacteria (Phylum-Bacteroidetes) regulates mucosal tolerance to self-antigens by maintaining T-cell homeostasis, preventing T-helper-1 (Th1)/Th17 balance [23] and induces suppressive forkhead box p3^+^ (Foxp3^+^) T-regulatory cells (Treg) functions by encouraging anti-inflammatory cytokines such as interleukin-10 (IL-10) and transforming growth factor-β (TGF-β) [24]. Similar preferment of Treg differentiation has been shown with the presence of *Clostridia* sp. in the colon, but not in the small intestine in maintaining the immune cell homeostasis [25]. Additionally, differentiation of Th17 cells and mucosal immunoglobulin-A (IgA) secretion are possible only with the colonization of segmented filamentous bacteria (SFB) [26]. Several mucosal pathogens such as bacteria, virus and fungi uphold both Th17 and IgA. The ecology of the microbiome is crucial, attaining the highest efficacy of vaccination since infants born because the immunization process has mostly been accomplished during childhood (Figure 1). An infant’s microbiome is determined by the maternal–offspring exchange of microbiota [27]. It becomes similar to the adult microbiome by three years of age. Until then, it is highly variable [28] and grows uninterrupted, along with the development of the immune system [20]. The total microbiome composition of infants is mostly occupied by the phyla such as Firmicutes, Tenericutes, Proteobacteria, Bacteroidetes and Fusobacteria based on commensal microbiota harbors in the placenta [29,30,31]. The variation of commensals is also related to mode of infant delivery. The intestinal flora of neonates born by vaginal delivery resemble the maternal vaginal flora. They are prevalent with *Lactobacillus* sp. and *Prevotella* sp. [32]. In another report, vaginal delivery supports higher existence of facultative anaerobic organisms like *Escherichia coli*, *Staphylococcus* sp., *Streptococcus* sp. and other Enterobacteriaceae (Phylum Proteobacteria) for first few days of birth [27,33]. Nonetheless, caesarean babies are predominantly colonized with maternal-skin microbiota such as *Clostridium*, *Staphylococcus*, *Propionibacterium* and *Corynebacterium* [34,35]. After birth, milk feeding has the most significant impact on changing the microbiome composition of children. Breast milk provides increased prevalence of *Bifidobacterium* sp. (phylum Actinobacteria) in the intestinal track of infants. Infants are also occupied with higher abundance of *Staphylococcus* sp. *Streptococcus* sp., *Lactobacillus* sp. (phylum Firmicutes), *Serratia* sp., *Ralstonia* sp. (phylum Proteobacteria), *Corynebacterium* sp. (phylum Actinobacteria), etc. in their gut [36]. In contrast, there is a richness of aerobic bacteria and lower frequencies of *Bifidobacterium* sp. in the guts of bottle milk-fed infants [37,38,39]. The neonatal gut microbiota not only programs the metabolic function, but also educates the naïve immune system, without which, the success of vaccination will be precluded.

### 2.2. Impact of Nutrients

The balance among nutrient metabolism, microbiota and vaccination efficiency is affected by leptin signaling [40], which regulates the hunger threshold, as well as cellular immunity by maintaining Th1/Th2 balance [41] and suppressing Treg cell differentiation [42]. In addition, GF mice have reduced leptin expression, suggesting a connection between microbiota and leptin-mediated immune response in the vaccination process [43]. Furthermore, several metabolites are critical, as short-chain fatty acids (SCFAs) show an example of how nutrient processing by microbiota shapes the development of the immune system [44]. SCFAs are generated exclusively by microbiota. Important SCFAs (butyrate and acetate) help to maintain intestinal epithelial integrity [45,46]. Vitamin A (retinoic acid) deficiency in the diet has also been linked to the amplified frequency of *E. coli*, causing enteric infections [47]. Vitamin A can promote the balance between both Treg and Th17 subsets. Imbalance of Vitamin A causes amelioration of Th17 subsets in the small intestine, which is associated with increased SFB [48,49]. Thus, microbiota frequencies are the indispensable factors for successful immunization. The impact of nutrients to support the microbiota symbiosis cannot be disregarded (Figure 1).

### 2.3. Impact of Environmental Factors

The aforementioned reports suggest that both nutrition metabolism and composition of microbiota influence the expansion of innate and adaptive immune systems (Figure 1). According to the Centers for Disease Control, every 1 in 10 children in the USA and one in four children in Europe suffer from various allergic disorders [50], bringing allergic diseases to an epidemic condition [51,52]. This is strongly aggravated in urban environments involving various pollutants. Exposure to various pollutants disturbs the frequencies of the microbiome in the body. In contrast, the concept of the hygienic hypothesis recommends that exposure to certain viral infections in early life and large family size reduces the risk of suffering from hay fever and allergic rhinitis [53]. The hygienic hypothesis basically takes into account the “critical period of development” or “window of opportunities” in early childhood, during which the immune system of the adult is shaped by intrinsic or extrinsic factors [54] (Figure 1). The shaping of ideal immune system decides the success of the vaccination. Dietary changes and the environmental factors can account for up to 57% of gut microbiota changes, affecting the immune system [55]. Consequently, westerners have less microbial diversity in the gut because of diets high in saturated fats and low in fiber, which affect the microbiota enrichments [56,57]. In a similar study, Filippo C et al. 2010, observed that the microbiota of rural Africans fluctuates vividly from that of city-dwelling European children. African children had more anti-inflammatory commensal bacterium such as *Bacteroides* sp., *Prevotella* sp., *Faecalibacterium prausnitzii*, *Xylanibacter* sp., etc. They had lower frequencies of both Firmicutes and Enterobacteriaceae that help in the generation of short-chain fatty acids (SCFA) for providing anti-inflammatory responses in the gut. These studies further implied that higher frequencies of anti-inflammatory commensal microbiota augmented the immunization adeptness to protect children from certain pathogens (including enteropathogens). In contrast, malnutrition aggravated vaccine efficacy in a community where microbiome dysbiosis is prevailing due to EE [12,58].

## 3. Route of Immunization Determines the Efficacy of Herd Immunity

The vaccine is the imperative prophylactic intervention preventing the burden of several infectious diseases [59,60]. Oral vaccines are used mostly in developing countries, although significant discrepancies exist in the competency of the oral vaccines based on the geographical distributions [61,62,63]. Many vaccines with both living or non-living agents, such as B-subunit-inactivated whole-cell combination vaccine [64], poliovirus vaccine, SC602 live *Shigella flexneri* 2a vaccine [65], oral rotavirus vaccine (RVV) [66], CVD 103-HgR live cholera vaccine [67,68], have also demonstrated differential immunogenic responses based on their route of immunization. In the case of oral poliovirus vaccine (OPV), live attenuated poliovirus is given orally in India from birth. OPV can spread efficiently and triggers the innate immunity followed by T-cell and B-cell activation because live microbes can persist for longer periods than dead organisms. Thus, significant T-memory cells and B-memory cells are generated, which are key factors for the competence of herd immunity. OPV can generate antibodies which limit the proliferation of challenged wild poliovirus and thwart person-to-person transmission. Nonetheless, in the USA, formalin-fixed inactivated poliovirus vaccine (IPV) is given in newborn babies. IPV cannot spread to different organs, and can activate only B-cell for antibody production. The dead vaccines sometimes escape T-cell activation because the antigens do not persist in the immune system for a substantial period. To activate T-cells, vaccine candidates must endure in the body long enough for antigen presentation (Figure 2a,b). In the case of IPVs, dead vaccine candidates are directly processed by B-cells to produce antibodies for neutralizing the pathogens. Thus, IPV poses only individual protection but cannot prevent the spread of wild poliovirus in the community. Hence, the chances of herd immunity against the IPV is very low. In different oral vaccinations, the microbiome plays a pivotal role in providing mucosal immunity [69]. The enhancement of mucosal immunity is the symphony of the immunization process. Similarly, in OPV, the intestinal microbiome has a significant influence on mucosal immunity-mediated adequate vaccine responses. This is deficient in IPV, in which the route of immunization is parenteral [70]. Thus, the microbiome seems to give plausible justification for developing herd immunity in the OPV (Figure 3).

## 4. Potential Role of the Microbiome in the Individual Vaccination to Community Immunity

### 4.1. Differential Expression of Microbiota Decides the Fate of Vaccination

Vaccines are the most important preventive interventions for reducing the burden of several infectious diseases [59,60]. The microbiome plays an essential role in shaping the immune system, and determines the efficacy of different vaccines, developing protective immune responses against variety of diseases. Earlier reports have suggested that differences in even normal residential intestinal microbiota between different individuals of a population can produce remarkable variations in efficacy of many vaccines, both locally and systemically [71]. The microbiota seem to give a plausible explanation for this (Table 1). Due to intestinal dysbiosis, there are different residential intestinal microbiota among different individuals. It has been shown that gut colonization of GF mice with human *Bifidobacterium* sp. led to enriching immune response against rotaviruses by increasing anti-rotavirus IgA secretion. Thus, *Bifidobacterium* sp. engrossment is required for the priming of the infant’s adaptive immunity [61,72,73,74,75]. Conversely, a lower vaccine response has been correlated with increased frequencies of “Enterobacteriales”, “Pseudomonadales” and “Clostridiales” in the case of both oral and parenteral vaccines [61]. This differential immunization response decreases the likelihood of developing community immunity in a particular cohort. In a separate study, a positive correlation was found between the intestinal *Bifidobacterium* sp. (*B. longum*, *B. infantis* or *B. breve*) and heightened anti-poliovirus IgA antibodies after immunization of OPV with a pentavalent diphtheria–tetanus–acellular pertussis–inactivated poliomyelitis–*Haemophilus influenzae* type B vaccine [76]. Thus, some specific microbiome frequencies are more desirable because they are a source of natural adjuvant for facilitating the efficacy of vaccine responses and developing herd immunity against pathogens [77].

### 4.2. Effect of EE on Vaccination

Metabolic dysfunction is associated with a weak response to many established oral vaccines. Due to low-incomes, the availability of a balanced diet in some populations is compromised. This exacerbates the efficacy of the oral vaccines such as cholera, poliovirus and rotavirus vaccines. Imbalance of nutrients cultivates EE, leading to small bowel bacterial overgrowth (SBBO) among some individuals [78]. Underprivileged children in slum communities commonly have excessive bacterial colonization at their proximal small intestine. This impairs the architecture of intestine with blunted villi, abnormal crypt-to-villus ratio [71,79,80,81]. Overburden of bacterial growth augments the number of intraepithelial lymphocytes [82,83], resulting in a marked increase in lymphocytes, which are the signature factors of allergic diseases [84,85,86]. Villous atrophy due to regular ingestion of both contaminated food and water containing fecal–oral bacteria due to poor sanitation aggravates plasmacytic infiltration in the lamina propria of the intestinal mucosa [87]. Concurrent enteric infections with entero-pathogens cause intensified small intestinal inflammation, resulting in intestine–barrier dysfunction and reducing the absorption of nutrients [88]. The aforementioned phenomenon can result in a blunted immunization process, as has been observed with live cholera vaccine CVD-103-HgR in an underdeveloped community. High seroconversion rates in Indonesian children living in poor conditions required 5 × 10^9^ colony forming unit (CFU) which is 10-fold higher dose of CVD-103-HgR than the 5 × 10^8^ CFU dose that is reliably immunogenic in North Americans and Europeans [67,68,69,70,71,72,73,74,75,76,77,78,79,80,81,82,83,84,85,86,87,88,89] who generally have healthy diets. Thus, the success of oral vaccines depends upon both diet and hygienic conditions. People with SBBO typically have immunologically activated small intestine [87], indicating a pro-inflammatory state in the small intestine. Thus, for any live or attenuated vaccine material, the proximal small intestine micro-environment becomes hostile, hampering the vaccine program. Therefore, due to poor induction of specific innate and adaptive immune responses in such populations, the chances of herd immunity are futile [90]. Live vaccines may then, instead of activating the innate immunity to enhance adaptive immune responses, be destroyed by already highly activated leaky innate immune-mediated inflammatory responses [91,92,93]. The low immunogenicity of oral vaccines due to EE aggravates herd immunity [60]. Similarly, the live oral *Shigella flexneri* 2a candidate SC602 vaccine also showed a protective response in North American volunteers with strong immune response [94]. In contrast, Bangladeshi toddlers who had poor diets did not have any immunized efficiency following the ingestion of variable CFU of SC602 [95]. In another study, infants from the slum area of Bangladesh showed reduced efficacy of both oral RVV and OPV due to intestinal injury, while there was no impact on the parenterally administered vaccines such as for tetanus, pertussis, diphtheria, *Haemophilus influenzae* type B and measles [96,97]. Because microbiome dysbiosis induced nutritional imbalance, the slum area of Bangladesh witnessed dominancy of both *Campylobacter* sp. and enterovirus in the gut. This was negatively correlated with the immunogenicity of OPV and also diminished IgA titer of rotavirus (Table 1) [98].

### 4.3. Cardinal Effect of Probiotics and Antibiotics on Avidity of Vaccination

Probiotics are defined as live microorganisms that, when consumed orally in adequate amounts, are beneficial to the vaccine efficacy of the host [71,102]. The advantageous effects of probiotics have been found to be the strongest in oral as well as parenteral influenza vaccine. Furthermore, the administration of a prebiotic-like fructo-oligosaccharide/inulin mix augmented efficacy of *S. typhimurium* SL1479 vaccination in a murine model [101], suggesting that resident gastrointestinal microbiota can be modulated by prebiotics and may confer improved immunological response towards vaccines.

Further studies have shown that rigorous exposure of antibiotics can exacerbate microbiome dysbiosis, impairing vaccine responses and may produce long lasting deleterious effects [84]. Infants exposed to antibiotics early in life have been shown to be more prone to diseases such as obesity [103], asthma [104] and metabolic syndrome [105]. Lynn et al. 2018 reported that antibiotic-mediated dysbiosis in early life leads to impaired antibody response against several vaccines. Thus, both probiotics and microbiome symbiosis are essential factors to sustain the long-lasting success of vaccines (Table 1).

## 5. Innate Immunity Controls Antibody Titter

The first line of defense in the immune system is innate immunity, which comprises different pathogen recognition receptors (PRRs). PRRs can recognize both pathogen-associated molecular patterns (PAMPs) of microbes and danger-associated molecular patterns (DAMPs), the cellular products produced in response to cell stress. Examples of PRRs include C-reactive protein, different TLRs, C-type lectin, nucleotide binding oligomerization domain like receptor (NOD), retinoic acid-inducible gene I (RIG-I), melanoma differentiation-associated gene 5 (MDA5), stimulator of interferon gene (STING), etc. [106,107] (Figure 4). Pathogen recognition is a critical practice that must efficiently differentiate the self from non-self to facilitate a specific immune response against microbial pathogens and to circumvent collateral damage rendered by autoimmunity. For RNA viruses, the RNA genome, its replication and metabolic products represent a major non-self-products harboring PAMPs that are recognized by several PRRs like TLR3 or RIG-I, MDA5 recognition [108] (Figure 4).

## 6. The Specificity of Herd Immunity Relies on Innate Immunity

Under the ideal infectious stage, different PAMPs of microbes activate the innate immune system (Figure 4) followed by both the arms of the adaptive immunity to generate the memory response (Figure 2) [109]. In case of herd immunity, this phenomenon happens to be in the mass fraction of the population to develop the memory response against the different antigens. Nevertheless, each antigen type has both benefits and difficulties which can perturb the activation of the immune system and limit the chances of developing herd immunity in the community [110]. In the case of a pandemic or epidemic situation, pathogens enter the body and the naked DNA/RNA of the microbes start to produce antigens. These microbial antigens are processed by the innate immunity displaying it on intracellular PRRs, e.g., TLR3, MDA5, STING, etc. (Figure 4) [106]. Activated innate immunity is involved in T-cells differentiation through APCs, developing memory T-cells (Figure 2a) and is followed by the activation of B-cells (Figure 2b). Activated plasma cells produce both a strong antibody response to neutralize the antigen and also memory B-cells, which provide protection against the same or similar types of infection in the future (Figure 2b) [111]. These memory responses act as a natural vaccination and develop herd immunity against the microbes during a pandemic or epidemic or endemic (repeated infection by similar pathogens in an area) situation. Antibody titer is critical for long-term herd immunity. Nakaya et al. 2011 completed a systematic biological analysis of trivalent-inactivated influenza vaccine (TIV) to investigate the antibody-generation sequence. A positive correlation was found between early expression of TLR5—an important PRR for bacterial flagella of the microbiome population—and magnitude of antibody response [99,112]. Consistent with this finding, it was also found that both GF and antibiotic-treated mice were unable to provide immunity to TIV, which suggests that the intestinal microbiota were a source of TLR5, helping in providing immunity against TIV. Nonetheless, the antibody titer was reinstated by oral reconstitution with flagellated *E. coli*. This observation also reveals a prominent role of the intestinal microbiota in controlling immunity to parenteral vaccines [100]. Several earlier reports have also shown that abrogation of gut microbiota symbiosis exacerbates the host susceptibility to several viral infections (Figure 3) [113].

## 7. Realm of Herd Immunity Can Be Established by Heterologous Immunity

The most important consequence of herd immunity is the establishment of persistent immunological memory responses. Homologous memory responses can be generated by encountering the pathogens by the innate immune system, followed by activation of adaptive-immune responses. Nonetheless, herd immunity can also be initiated upon exposure to different pathogens or antigens in a particular cohort, known as heterologous immunity [114]. Conceptually, the cross-reactivity (or poly-specificity) of lymphocytes in antigen (or epitope) recognition is the fundamental concept of heterologous herd immunity (Figure 5a). Cross-reactive memory-T-cell responses, followed by memory-B cells responses are engaged at the evolutionarily conserved sites among several virus groups, such as different strains of influenza or Dengue virus (DENV) or among different members of the same virus group, such as, arenaviruses, flaviviruses and hantaviruses [115]. Nonetheless, the examples of cross-reactive T-cell or B-cell responses in heterologous immunity involving absolutely isolated viruses have also been found between human papillomavirus and coronavirus [116] or influenza virus and hepatitis C virus (HCV) [117] or influenza virus and HIV [118] or lymphocytic choriomeningitis virus (LCMV) and vaccinia virus (VV) [119,120] or influenza virus and epstein–barr virus (EBV) [121].

In these infections, cross-reactive antibodies-mediated responses are frequently observed due to the presence of the heterologous, but comparable antigens. After immunization with different vaccines, the body has a mixture of different antibodies. Among these various antibodies, one or more can recognize the antigens of second pathogens (Figure 5a). This heterologous immunity is reasonably supportive to engender herd immunity against the new pathogen in a community [122,123]. Nonetheless, heterologous herd immunity can be demonstrated by “coinfections” where the two unrelated pathogens infect simultaneously or within a short window period prior to the systemic dissemination of the first pathogen in the host or “superinfections” where a second pathogen enters after the first pathogen is well-established.

## 8. Heterologous Immunity Provides Significant Protection against SARS-CoV-2 Infection in India

### 8.1. Herd Immunity against Coronavirus and Impact of Microbiota

The generation of herd immunity against coronavirus still has a long way to go. The precise herd immunity threshold for the SARS-CoV-2 is not yet clear. However, several experts believe that if the spread of infection is higher than 60% in a population, then herd immunity can be developed against this virus [11]. Several studies have observed, however, that the size of the tested population is broad, but the percentage of people who have been infected so far is still in a single digit. This result is due to stringent lockdowns in different parts of the world. Several countries—notably Sweden and briefly Britain—have experimented with limited lockdowns in an effort to build up herd immunity in their populations. However, even in these places, recent studies indicate that no more than 7–17% of people have so far been infected. In the month of May 2020, SARS-CoV-2 infection spread to as many as 20% of the total city dwellers in New York City, USA. This is the highest number for an outbreak in a particular cohort in USA. Both the density of city dwellers in an area and the physical contacts among individuals are critical factors in reaching the threshold of herd immunity [124,125]. Based on several experts, on average 60% of the population must come in contact with the pathogen or its antigens to achieve herd immunity. Thus, with the possibility of a faster spread of diseases than is currently believed, the herd immunity generation could be faster. The variation of immunity, nutrition and epigenetic effect among residents is likely to drive herd immunity generation, even downwards [126]. Several previously common infectious diseases among children, e.g., measles and chickenpox, are now extremely sporadic in the United States because vaccines have helped to create enough herd immunity to impede the outbreaks [2,127]. COVID-19 is potentially a much more dangerous disease than any other pneumonia-causing pathogens. This virus infects mostly those who are at risk of getting sick due to carrying any other disease from the past. In several countries, e.g., USA, many people already have threshold immunity, either because they had been sick with a similar viral strain of pneumonia in the past, or because they have received a shot of variable vaccines. Among these vaccines, one or few may have a good match for the version of the virus they encountered. This number is not high enough for reaching herd immunity—and those viruses still circulate in every year. However, there are benefits of having partial immunity in a population, which helps to lower the risk of death in that particular population. COVID-19, unlike influenza, is a brand-new disease. It has the potential to kill many more people due to SARS-CoV-2-antigen-specific poor immunity in a population.

SARS-CoV-2 is certainly the cause of a pandemic, but also a potential solution against infection by developing herd immunity. It is imperative to investigate the impact of microbiome in cultivating herd immunity against SARS-CoV-2. Virulence factors of SARS-CoV-2 provoke the hyper-inflammatory response generated by the body’s immune system—also called cytokine storm syndrome—which is responsible for many complications and deaths [128]. The gut microbiome is the foremost factor to control the cytokine storm [129]. Inflammatory responses are controlled by specific microbiota composition, which may predict predisposition of COVID-19 [130] (Table 2). MRx-4DP0004, a strain of the bacterium *Bifidobacterium breve* in the phylum of Actinobacteria, originally developed for asthma [131], inhibited hyper-inflammatory response by reducing the expression of angiotensin-converting enzyme 2 (ACE2) receptors for maintaining the potential antiviral response [132]. The prevalent of the SARS-CoV-2 receptor, ACE2, is highly expressed in microbiome-enriched gut enterocytes and colonocytes [70,133]. Thereby, in the current crisis, the impact of microbiome cannot be ignored. The “microbial dark matter” inside the body and around us indeed endorses the potential to provide us with the tools to develop immunity with a limited spread of multidrug-resistant pathogens.

### 8.2. Microbiome: New Songs in Old Music

Microbiome frequencies between lung–gut axis are the critical and comprehensive biomarkers for viral diseases. Several studies revealed that higher frequencies of *Capnocytophaga gingivalis*, *Veillonella* sp., *Leptotrichia buccalis*, *Veillonella parvula* and *Prevotella melaninogenica* in the bronchoalveolar lavage fluid (BALF) exacerbated the COVID-19 patients [134]. These microbiome populations augmented the synthesis of both nucleotide and amino acids and enhanced the metabolism rates of carbohydrates, which is precursor of a worsening infection. The respiratory tract and lungs are enriched with *Streptococcus infantis* during SARS-CoV-2 infection [135]. Conversely, *Fusobacterium periodonticum*, a prevalent component of the of the lung microbiota has prohibited the severity of COVID-19 [136]. The imbalance of microbiota frequencies in gut also increase the severity of pathogenesis in SARS-CoV-2 infection due to variable expression of ACE2. The few bacteria of microbiome population help in maintaining the ACE2 expression which expedite the infection rate in the host. *Collinsella aerofaciens*, *Collinsella tanakaei*, *Streptococcus infantis*, *Morganella morganii* are preeminent commensals in gut microbiome of severely infected COVID-19 patients [135]. Several opportunistic pathogens have been persistently associated with exaggerated patients with COVID-19 (e.g., *Ruminococcus gnavus*, *Clostridium hathewayi*, *Enterococcus avium*, *Collinsella aerofaciens* and *Morganella morganii*). In contrast, a few bacteria such as *Parabacteroides merdae*, *Bacteroides stercoris*, *Alistipes onderdonkii* and *Lachnospiraceae bacterium* of intestinal microbiota have the competence to reduce ACE2 expression in the gut epithelial cells, thereby influencing the calibrated immune system to encumber the pathogenicity of COVID-19. These commensal bacteria sustain the nutritional factors in gut. It is cardinal for enhancing the immune system and essential for the protection against the SARS-CoV-2 infection. Short-chain fatty acids (butyrate, acetate) are synthesized by *Parabacteroides merdae*, *Bacteroides stercoris*, *Alistipes onderdonkii* and a Lachnospiraceae bacterium; an abundance of these bacteria is imperative for providing prophylactic efficacy against COVID-19. Tryptophan metabolism to melatonin by *Alistipes onderdonkii* is necessary to reduce the infection. These bacteria have a salutary role in combating to SARS-CoV-2 infection [137,138] (Table 2). Other than bacteria, COVID-19 patients also have higher frequencies of *Aspergillus flavus*, *Candida glabrata* and *Candida albicans* which are also pertinent commensals of the human gut microbiome [139].

### 8.3. Herd Immunity: Friends or Fes for Covid-19

A wide variety of human leukocyte antigen (HLA) molecules ensures that individuals across the population present diversified antigenic peptides. It ensures the utmost chance of individuals of that particular population may survive against the emerging diseases. In contrast, a ‘footprint’ of immune responses is established during the first exposure of a pathogen. Re-exposure of the same or a similar type of antigen of latter pathogen stimulates and cross-reacts with memory T-cells which are specific for antigens of former pathogen or vaccination (Figure 2a). Thereby, an immune repertoire memory T-cells have been preferentially re-expanded which refuted the clonal expansion of new antigen-specific T-cells. Consequently, the chances of generation of a new type of memory T-cells of latter pathogen will be thwarted unwittingly, known as ‘original antigenic sin’ (Figure 5b) [140]. In that case of heterologous herd immunity, T-cell cross-reactivity and the HLA diversity to past pandemics can encourage suboptimal protective immune response to the second pathogen owing to ‘original antigenic sin’ (Figure 5b). A similar type of mechanism has been proposed for B-cells responses (Figure 2b). This antigenic sin can be extended beyond a simple case of low sensitivity to the second antigen to an inferior situation in which the original antigen has established a T helper 1 (Th1)- Th2- or Th17-type of responses, which are unsuitable as well as attrition for the second pathogen. This dampens the concept of heterologous herd immunity development which is required for evolving the community vaccination [141].

### 8.4. Heterologous Immunity: Poliovirus Vaccine May Provide Protection against SARS-CoV-2 Infection

In a pandemic situation, the entire world has a severe infection with SARS-CoV-2. To date, there is a very low rate of death of infected children under age of 10, as well as for those not having a history of any persisting disease (like any type of autoimmunity). This is a significant observation that children who have received regular immunization of different vaccines may have heterologous protection against this COVID-19. In pan-India, the continuum of mass vaccinations with OPV, BCG, measles vaccine, etc. has established memory responses against many types of variable antigens in the body. A single-type or multiple-types of antibodies from the pool of the antibody mixture, present in the body, can recognize the novel pathogens carrying the same or similar types of antigens. This phenomenon is considered the major reason for the unexpectedly low rate of COVID-19 occurrence in India with a faster recovery rate. The infection rate until September 15, 2020, with SARS-CoV-2 is only 8.45% of tested population. The total tested samples are almost 58 million, whereas the sum of the total infected patients including death, recovered and active is 4.9 million in India. If we compare the total infected samples (4.9 million) with the total population of India (1300 million), then the percentage of infected patients is very negligible (0.37%) which validates that SARS-CoV-2 infection has been cleared by the community comfortably. The total mortality—including co-morbidity deaths due to COVID-19—is 80,808, which is actually 0.006% of the pan-India population (1300 million). This higher percentage of protection has been possible due to heterologous memory immune response against COVID-19. Dominant antibodies, generated from previous vaccinations have the aptitude to clear the SARS-CoV-2 comfortably, thus the possibilities of developing herd immunity against recessive antigens of SARS-CoV-2 are essentially very subdued. Innate and adaptive memory responses in the course of poliovirus vaccination collude to develop heterologous immunity against SARS-CoV-2 infection and are helping to slow down the spread of COVID-19 [142]. Among different vaccines, two candidate vaccines are of paramount importance and are also gradually becoming the focus of repurposing of vaccines for COVID-19: the poliovirus vaccine and the BCG vaccine [142,143]. The BCG vaccine originally administered to provide protection against *Mycobacterium tuberculosis* (TB) infection. BCG also has been shown to provide heterologous protection against other un-related infections also, as in yellow fever viral infection by inducing epigenetic reprogramming of monocytes [144] and reducing the child mortality rate [145]. BCG also showed heterologous immunological effects in low-birth weight infants [146]. There has been comparatively less number of SARS-CoV-2 infection-related deaths observed in countries who have continued with the usage of OPV than those countries who have been switched to IPV [143]. This correlation of OPV usage and the subdued COVID-19 mortality rate is urging the scientists to think to explore the possibility of poliovirus vaccine to help in lowering the COVID-19 severity. Furthermore, the similarity of SARS-CoV-2 encoded 3C-like protease and 3CPro of picornaviruses which has the role in viral pathogenesis through viral protein maturation, has also been used to design drugs against the 3C-like protease of SARS-CoV-2 to impede the infection [147]. These available vaccines can provide protection against COVID-19 possibly by antigen similarity with SARS-CoV-2, or through antigen-independent innate and adaptive memory [148]. We here tried to look at the first possibility discussed in the next section.

### 8.5. In Silico Comparison of SARS-CoV-2 with OPV and BCG

To explore the chances of antigen sharing, we ran a preliminary analysis, comparing the SARS-CoV-2 epitopes with OPV and BCG epitopes at the sequence level using protein basic local alignment search tool (BLAST). For this comparison, the information of efficient epitopes of the three organisms was extracted from the Immune Epitope Database and Analysis Resource (IEDB) [149] which is a database for experimentally known epitopes. We downloaded the sequences of SARS-CoV-2 (ID: 2697049), all three types of poliovirus (ID: 12080,12083, 12086) and BCG (ID: 33892) with only “linear epitopes” and “human host” using both as the filters from IEDB which provided 321, 88 and 464 epitope sequences for the three respective organisms. As in the majority of epitopes, these epitopes were 9–12 amino acids long. Then, Protein BLAST was run for each of these 321 SARS-CoV-2 epitope protein sequences against the database of downloaded Polio and BCG epitope sequences using the national center for biotechnology information (NCBI) offline Protein BLAST tool [150]. Considering the usual length range of epitopes (9–12 residues), an E-value cutoff of 10e-2 and minimum alignment length of eight residues with no gaps was used to filter the BLAST hits. Upon performing this analysis, SARS-CoV-2 open reading frame 7a (Orf7a) protein epitope (residue 72 to 81 with sequence ‘KHVYQLRARS’) showed similarity (80%) with human poliovirus type 3 Sabin strain epitope (residue 62 to 71 with sequence ‘RHVVQRRSRS’) of VP1 protein. Similarly, the SARS-CoV-2 nucleocapsid epitope (residue 201 to 209 with sequence ‘SSRGTSPAR’) shared similarity (87%) with part of human poliovirus type 1 Mahoney epitope (residue 186 to 193 with sequence ‘TYGTAPAR). No such direct matching of epitope sequence was found between BCG and SARS-CoV-2. However, this is only the first step to analyze antigen similarity and requires other important factors such as 3D structure of epitopes, their subcellular localization, anti-sera profile, etc. to be taken into account to understand the complete scenario of cross-reactivity of antibodies.

## 9. Perspective and Future Opportunity

The concept of herd immunity—or community immunity—has triggered an intense debate in concerning whether it would prevent the pandemic of COVID-19, and if so, how much herd immunity would be required to effectively impede its spread? Along with this, an important unanswered question in the current pandemic situation is the discovery of a future probable vaccine against SARS-CoV-2, against which the propensity of herd immunity expansion is rather uncertain. Based on theory of original antigenic sin, the second infection of an unlike pathogen activates the memory immune response that was developed from the first infection. The first pathogen-specific antibodies can neutralize new dominant antigens of the second pathogen. The effective antibodies against those dominant antigens coming from first infections, are thus able to eradicate the second infection. The immune system needs a reasonable time to recognize the new antigens through APCs, followed by T-cell activation, generation of memory T-cells and triggering of B-cells for antibody secretion. Nonetheless, if dominant antibodies encounter comparable antigens from the second pathogen that are enough to eliminate the later infection, then the rest of the different antigens of second pathogen will be presumed to be recessive antigens. The immune system will only generate the former dominant antibodies against this new second pathogen, declining to produce new antibodies against the first pathogen. This phenomenon results in an inefficient and a weak immunity, which has been expected against SARS-CoV-2 infection in India due to regular poliovirus vaccinations. Thus, the probability of attaining herd immunity against COVID-19 is very limited given the current context of its low infectivity rate.

## 10. Conclusions

Herd immunity is the most critical prophylactic intervention, delivering the protective immunity against several infectious diseases such as smallpox, poliovirus, measles, etc. in the past. The new paradigm of evolving herd immunity during a pandemic situation cannot be disregarded.Crosstalk among microbiota, metabolism and environmental factors is critical for developing a competent immune system, which is a prerequisite for evolving herd immunity against any contagious infections;The importance of herd immunity has been documented with the current context of a pandemic scenario due to transmissible infection of SARS-CoV-2 virus;In India, the infection rate of SARS-CoV-2 is unexpectedly very low, i.e., only 0.37% of total population. The prevalence of heterologous immunity due to rigorous vaccination programs at the grass-root level may provide protection against the SARS-CoV-2 pandemic;Comparisons between SARS-CoV-2 Orf7a protein epitope (KHVYQLRARS) and the human poliovirus type 3 Sabin strain epitope (RHVVQRRSRS) from VP1 protein offer a great insights into the concept of heterologous immunity, which can be an alternative providing prophylactic intervention against both the poliovirus and COVID-19.

## Figures and Tables

**Figure 1 viruses-12-01150-f001:**
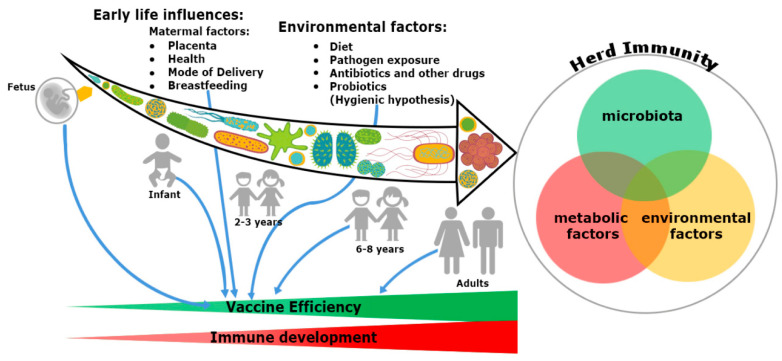
Trinity of the immune system development and vaccine efficiency. The immune system and microbiota mutually co-evolve together in a symbiotic relationship. The impact of microbiome on the immune system development cannot be ignored. From fetus to adulthood, microbiomes are synchronized by maternal transfer and environmental factors. Early maternal factors such as mode of delivery, breastfeeding, antibiotics and diets all influence the immune system. Hence, all have an impact on subsequent immunological responses to many vaccines. Development of herd immunity in a community against any infection is the result of a complex outcome of host-specific factors such as microbiota, metabolism and environmental conditions. Malnutrition and repeated gastrointestinal infections reduce many vaccines’ efficacy. Microbial dysbiosis, along with environmental enteropathy (EE) influences undernourishment. It impairs the immune system development and decreases the efficiency of vaccines in a community, thereby compromising herd immunity.

**Figure 2 viruses-12-01150-f002:**
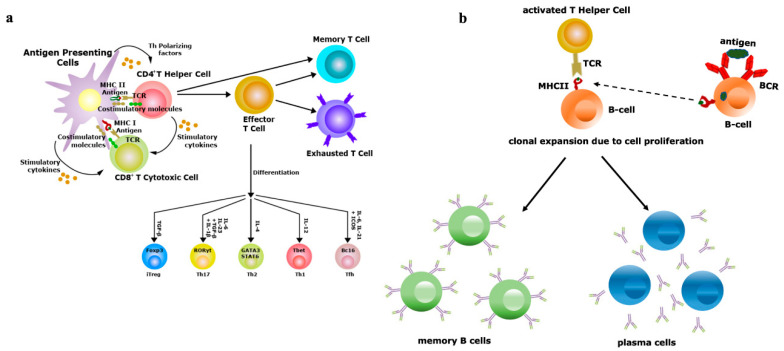
Vaccine efficiency depends on the repertoire diversity of T-cells and B-cells. (**a**) Upon contact with antigen-presenting cells (APCs) through the T-cell receptor (TCR)–antigen–major histocompatibility complex (MHC) complex, naïve T-cells are differentiated into effector and memory T-cells. Based on the antigenic property, effector T-cells are of two types, i.e., CD4+ T-helper (Th) or CD8+ T-cytotoxic (Tc) cells. Antigens exposed by MHCII binds with TCRs of Th cells and antigens exposed by MHCI binds with TCRs of Tc cells. Th cells have the potential to further differentiate into Th1, Th2, Th17 and induced regulatory T-cells (iTreg), a process controlled by the lineage-specific transcription factors and effector cytokines produced by APCs. Effectiveness of memory T-cells decides the efficacy of vaccination; (**b**) B-cells (APCs) recognize antigens by their B-cell receptors (BCRs), followed by internalization of antigens and presented to Th cells, which are specific to same antigen. The TCRs of Th cells interact with antigens exposed by MHCII of B-cells. Then, activated B-cells trigger their own proliferation and differentiate into antibody-secreting plasma cells and memory B-cells. Activation and class switching of B-cells is the key factor for the success of immunization.

**Figure 3 viruses-12-01150-f003:**
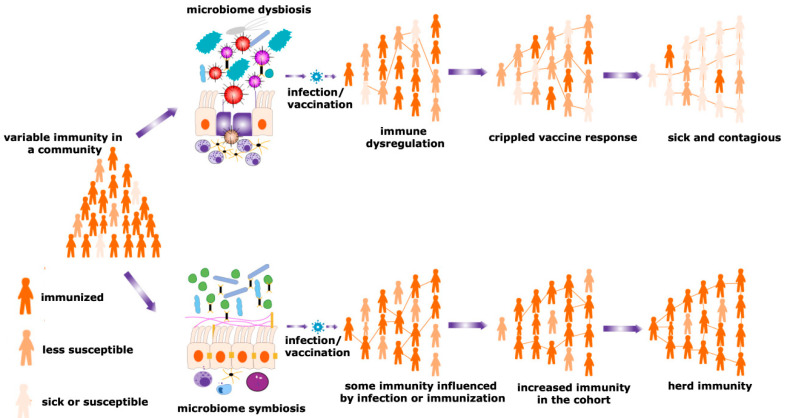
Herd immunity and microbiota. Maturity of the herd immunity against any infectious disease is a significant means of acquiring protection during an epidemic or pandemic situations. A spatial community is occupied by people having a variable degree of resistance against a particular infection. Microbial symbiosis regulates pathogen transmission dynamics and the efficacy of vaccination among different individuals in a population, promoting the development of herd immunity. In contrast, dysbiosis of the microbiome causes immune dysregulation, generating the suboptimal immune response and negative impacts on a vaccination program. Therefore, the number of sick individuals increases in a particular cohort. This reduces the chances of evolving herd immunity and increases the sick and contagious patients in a community.

**Figure 4 viruses-12-01150-f004:**
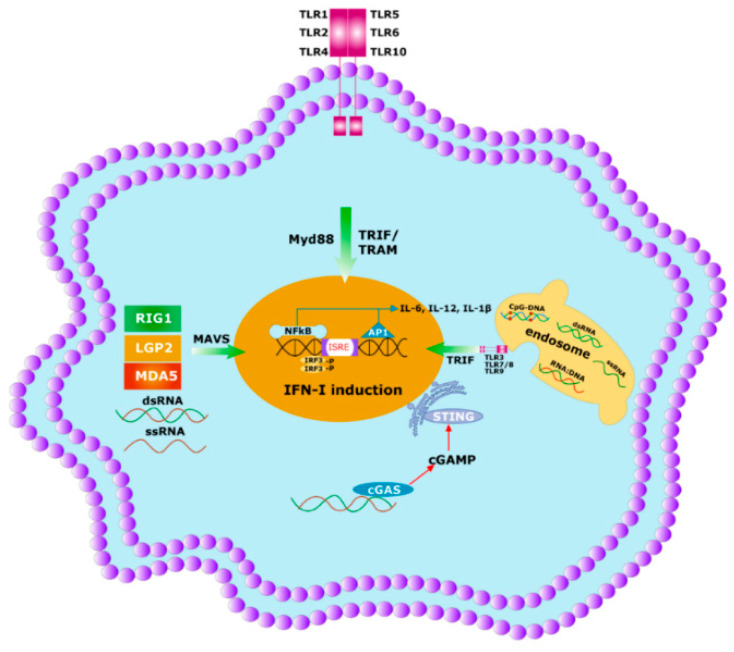
Initial immune response to any pathogen involves the activation of innate immunity. Innate immunity is activated by germline encoded PRRs which recognize PAMPs and DAMPs of the microbes. Specific PRRs mainly include membrane-bound several TLRs, cytosolic DNA sensor, e.g., STING, Rig -1 like receptor (RLR), MDA5, NOD-like receptor, all of which coordinate with the host innate immune responses through the activation of the nuclear factor κB (NF-κB), activator protein 1 (AP-1) and interferon regulatory factor (IRF)-signaling pathway which triggers the antiviral interferons type I (IFNI) pathway. Activated innate immunity can commence the adaptive immunity by triggering both the T-cells and B-cells.

**Figure 5 viruses-12-01150-f005:**
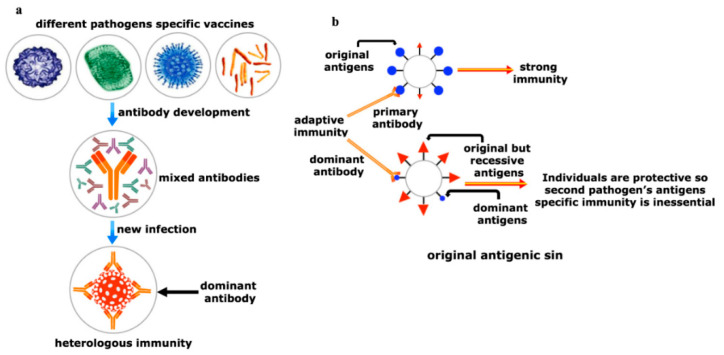
Yin and Yang of the heterologous immunity. (**a**) Heterologous immunity develops from the opinion where the antibody of one pathogen may protect an individual from a similar or in some cases even from phylogenetically distinct invaders. Vigorous immunization program with several vaccine candidates is required to improve protective response against any new incoming pathogen having comparable antigens. Both humoral and cellular cross-reactive heterologous memory responses may influence the foundation of the natural course of herd immunity against any new infection; (**b**) original antigenic sin refers to a phenomenon where the development of immunity against pathogens or antigens is shaped by the first exposure of related pathogens. The memory antibodies generated as a result of first exposure will interact with similar antigens of second pathogen for neutralizing it. This anamnestic recall against the second pathogen dampens the probability of immune system activation against original antigens of the second pathogen. This phenomenon is reasonably applicable to SARS-CoV-2 infection where its infection rate is very mild.

**Table 1 viruses-12-01150-t001:** Impact of intestinal microbiome on immune responses and vaccination.

1. Intestinal dysbiosis: Differences in microbiota composition of vaccine responders and nonresponders among different populations.		
**Vaccine Type**	**Microbial Richness Responsible for Vaccine Effectiveness**	**Tested Population (Ref)**
Pentavalent diphtheria–tetanus–acellular pertussis-inactivated poliomyelitis–*Haemophilus influenzae* type B vaccine (DTaP-IPV-Hib)	Existence of intestinal *Bifidobacterium* sp. especially *B. longum, B. infantis* and *B. breve* under phylum Actinobacteria enhanced anti-poliovirus IgA antibodies.	Infants (France) [76]
OPV, Parenteral tetanus toxoid (TT), *Bacillus* Calmette–Guérin (BCG) and hepatitis B vaccine (HBV)	Abundance of *Bifidobacterium longum* of phylum Actinobacteria augmented oral and parenteral vaccine avidities among infants. Amplified frequencies of *Enterobacteriales* and *Pseudomonadales* encumbered vaccine specific immune responses.	Infants (Bangladesh) [61]
RVV	i. *Streptococcus bovis* under Firmicutes correlated with oral RVV response. Increased *Bacteroides* sp. and *Prevotella* sp. of phylum Bacteroidetes were associated with lack of RVV response.	Infants (Ghana) [74]
	ii. Abundance of bacteria related to *Clostridium* cluster IX of phylum Firmicutes and *Serratia* sp. and *E. coli* of phylum Proteobacteria augmented RVV efficiency in a population.	Infants (Pakistan) [73]
Trivalent inactivated influenza vaccine (TIV) and OPV	Activation of toll-like receptor 5 (TLR5) enriched antibody response against TIV and OPV. Impaired antibody response in both GF mice and antibiotic-treated mice was restored by oral reconstitution with flagellated strain of *E. coli*.	Mice [99,100]
Live attenuated oral typhoid (Ty21a) vaccine	Cell-mediated immune response was found to be associated with SFB under phylum Firmicutes, while humoral response was independent of any microbial dysbiosis.	Healthy adults [72]
2. EE: SBBO disturbing the normal gut microbiota is responsible for suboptimal response of many vaccines in low-income settings and developing countries		
Live cholera vaccine CVD-103-HgR	SBBO could blunt the immunological response to live cholera vaccine candidate CVD 103-HgR	Children from high- and low-income countries [60,89]
Live oral *Shigella flexneri* 2a candidate SC602 vaccine	Lower dose of 10^4^ CFU was adequate to elicit a protective response in north American cohort while the ingestion of 10^4^, 10^5^ or 10^6^ CFU of SC602 was not adequate for Bangladeshi toddlers because of SBBO.	[94,95]
Oral RVV, OPV	i. EE was found to be associated with refuted efficacy of oral RVV but had no impact on the parenteral administered vaccines– tetanus, pertussis, diphtheria, *Haemophilus influenzae* type B and measles.	[96,97]
	ii. Enhanced frequencies of *Campylobacter* sp. under phylum Proteobacteria and enteroviruses at the time of immunization was negatively correlated with immunogenicity of OPV and diminished rotavirus immunoglobulin A titer (RVI).	Urban slum area of Bangladesh [98]
3. Probiotics and prebiotics: Several studies mentioned about the role of both probiotics and prebiotics on vaccine efficacy which further substantiates the significance of residential gut microbiota on vaccine effectiveness		
The impact of several probiotic strains on the efficacy of 17 different vaccines, e.g., diphtheria, tetanus toxoids and pertussis (DTP), whole-cell DTP vaccine (DTwP), diphtheria, tetanus, acellular pertussis, with *Haemophilus influenzae* type B (DTaP-Hib), DTaP-IPV-Hib, hepatitis A vaccine (HAV), hepatitis A vaccine (HBV), Hib, live attenuated influenza vaccine (LAIV), attenuated virus MMR vaccine with chickenpox vaccine or varicella vaccine (MMRV), oral cholera vaccine (OCV), OPV, oral rabies vaccines (ORV), pneumococcal conjugate vaccine (PCV7), pneumococcal polysaccharide vaccine (PPV23), polio, trivalent inactivated influenza (TIV) and Ty21a	*Lactobacillus* sp., *Bifidobacterium* sp. and *Saccharomyces boulardii* were the most frequently used microorganisms in probiotics. The beneficial effects of probiotics were found to be strongest in both oral and parenteral vaccines	[71]
*S. typhimurium* SL1479 vaccination	The administration of a prebiotic (a nondigestible food component that promotes the growth of beneficial microorganisms), e.g., fructo-oligosaccharide/inulin mix was shown to enhance efficacy of *S. Typhimurium* SL1479 vaccination	Mice [101]
4. Antibiotics: Several studies in both mice and humans have shown the effect of antibiotic-mediated dysbiosis on vaccine responses		
BCG vaccine, *Bexsero meningococcal* serogroup B vaccine (MenB), the meningococcal serogroup C vaccine-NeisVac-C (MenC); the Prevenar 13-valent pneumococcal conjugate vaccine (PVC13); the hexavalent combination vaccine against hepatitis B, diphtheria, tetanus, pertussis, *Hemophilus influenzae* type B, inactivated poliomyelitis virus (INFANRIX Hexa)	Antibiotic-mediated dysbiosis in early life impaired antibody response against these vaccines. Restoration of commensal microbiota retrieved vaccine efficiencies.	Newborn mice [13]
RVV	Positive correlation was observed in antibiotic-driven microbiota modulation and increased immunological response	Adult cohort [73]

**Table 2 viruses-12-01150-t002:** Differential expression of commensal microbiome between SARS-CoV-2 sensitive and resistance patients.

Alternation of Microbiome Frequencies Due to SARS-CoV-2 Infection
Sensitive Patients	Protective Patients
**Phylum Bacteroidetes**	**Phylum Bacteroidetes**
*Capnocytophaga gingivalis*	*Alistipes onderdonkii*
*Prevotella melaninogenica*	*Parabacteroides merdae*
*Bacteroides nordi*	*Bacteroides stercoris*
*Capnocytophaga* sp.	*Alistipes onderdonkii*
**Phylum Firmicutes**	*Bacteroides ovatus*
*Clostridium ramosum*	*Bacteroides dorei*
*Clostridium hathewayi*	*Bacteroides thetaiotaomicron*
*Erysipelotrichaceae bacterium*	*Bacteroides massiliensis*
*Ruthenibacterium lactatiformans*	*Bacteroides stercoris*
*Veillonella* sp.	**Phylum Firmicutes**
*Veillonella parvula*	*Faecalibacterium prausnitzii*
*Streptococcus infantis*	Lachnospiraceae bacteria
*Ruminococcus gnavus*	*Eubacterium rectale*
*Enterococcus avium*	*Ruminococcus obeum*
**Phylum Actinobacteria**	*Dorea formicigenerans*
*Actinomyces viscosus*	*Lactobacillus* sp.
*Corynebacterium* sp.	Lachnospiraceae bacteria
*Collinsella aerofaciens*	**Phylum Actinobacteria**
*Collinsella tanakaei*	*Bifidobacterium* sp.
**Phylum Fusobacteria**	**Phylum Fusobacteria**
*Leptotrichia buccalis*	*Fusobacterium periodonticum*
**Phylum Proteobacteria**	
*Acinetobacter baumannii*	
*Klebsiella pneumoniae*	
*Morganella morganii*

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
