# Peer review of "Impact of Microbiota: A Paradigm for Evolving Herd Immunity against Viral Diseases"

_viruses, 2020, doi:10.3390/v12101150_

Round 1

Reviewer 1 Report

This review discusses the well-timed topic, Impact of microbiota and its potential role in herd immunity against SARS-CoV-2 and other viral diseases. This review beautifully summarized a wide range of published work and discussed the potential role of the microbiome in herd immunity, prophylactic intervention, and protective immunity against the range of infectious diseases. The present review also discussed how cross-talk between environmental factors like microbiota and metabolism are important for the development capable immune system and herd immunity. Further, it has been highlighted that rigorous vaccination in India helpful in the development of heterogeneous immune response and protection against SARS-CoV-2 viral infection. It has been written in a clear and comprehensive way with good model figures, which will be highly helpful for the readers to understand the association among various factors (microbiota immune system and herd immunity). However, some revision may improve the quality of the review article. Author should consider adding a Table having details of recent report related to SARS-CoV-2 and other viral infections mediated changes in the abundance of specific bacteria. Furthermore, for reader convenience, authors should compile recent and earlier study in a table about the importance of microbiota in the herd immunity and vaccination success against viral diseases.

Specific Comments:

  1. For reader convenience, authors need to compile recent and earlier study in a table about the importance of microbiota in the herd immunity and vaccination success against viral diseases. Author may consider adding details of SARS-CoV-2 or other viral infections specific changes in abundance of specific bacteria. 
  2. It needs to be proofread for proper English grammar and small mistakes -a careful read through to correct these errors will significantly increase readability. Few Examples: Use of articles (a and the) are inconsistent throughout the manuscript  
    1. This process of natural vaccination is immensely pertinent to the current context of pandemic caused ….. context of “a” pandemic caused
    2. The conventional idea of herd immunity is established due to the efficient transmission of pathogens and developing the natural immunity in the populations…. Developing natural immunity in the populations.
    3. This led to achieve the herd immunity in a particular cohort. … This led to achieving herd immunity in a particular cohort
    4. The microbiome play an essential ……. The microbiome “plays” an essential
  1.  
  2. Some sentence needs to be re-written as the same is showing similarity with text from earlier published article:

“in the United States, around 20% of the city’s residents have been infected by the virus as of early May”,

The world is still far from herd immunity for coronavirus. https://www.msn.com/en-us/news/us/the-world-is-still-far-from-herd-immunity-for-coronavirus/ar-BB14JlG6

“The herd immunity threshold may differ from place to place, depending on factors like”

The World Is Still Far From Herd Immunity for Coronavirus .... https://www.nytimes.com/interactive/2020/05/28/upshot/coronavirus-herd-immunity.html

“could be higher. If there is a lot of variation in people’s likelihood of becoming infected when they are exposed, that could push the”

The World Is Still Far From Herd Immunity for Coronavirus .... https://www.nytimes.com/interactive/2020/05/28/upshot/coronavirus-herd-immunity.html

“composition may predict predisposition to severe COVID-19 due to hyper-inflammatory response [103]. MRx-4DP0004, a strain of the bacterium Bifidobacterium breve,”

Microbiome after coronavirus: Investing in the power of .... https://www.microbiometimes.com/microbiome-after-coronavirus-investing-in-the-power-of-microbes/

“The specific memory T cell populations are preferentially re-expanded when re-exposed to the same antigen or one that is similar, thereby limiting the clonal expansion of new antigen-specific T cells (Fig. 2a) [106]. A similar mechanism has been proposed for B cell responses (Fig. 2b).”

Why must T cells be cross-reactive? | Nature Reviews .... https://www.nature.com/articles/nri3279

  1. There is a lack of supporting data (i.e., bibliography) in some parts of the text. I would recommend the author reviewing the text and try to support some of the affirmations made. Reference needed for the following text
    1. Underprivileged children in the slam community commonly have excessive bacterial colonization at their proximal small intestine. It impairs the architecture of intestine with blunted villi, abnormal crypt to villus ratio.
    2. Overburden of bacterial growth augmented the number of intraepithelial lymphocytes and a marked increase in lymphocytes which are the signature factors of allergic diseases.
    3. Therefore, the chances of herd immunity are quite futile in that population due to poor induction of specific innate and adaptive immune responses.
    4. Live vaccines might then, instead of activating the innate immunity to enhance adaptive immune responses, be destroyed by an already highly activated leaky innate immune-mediated inflammatory responses.

Author Response

Reply to First reviewer’s concerns:

We, all authors sincerely appreciate the editor and all learned reviewers for giving us the opportunity to improve the quality of review article. We have worked on it and answered all the issues raised by reviewers.

We sincerely appreciate your kind suggestions.

Here is the point wise response to each question:

Major comments:

  1. For reader convenience, authors need to compile recent and earlier study in a table about the importance of microbiota in the herd immunity and vaccination success against viral diseases. 

Answer: Reviewer’s suggestion has been well taken, and accordingly modified in the manuscript.

Page 6-11: i. We have added an extra section (4.1. to 4.3.) which potentially depicting the role of microbiota, environmental enteropathy, probiotics-prebiotics and antibiotics for determining the efficiency of vaccination.

  1. In addition, we have added Table 1 (Page 8) which summarizes the role of microbiota, environmental enteropathy, probiotics-prebiotics and antibiotics on different types of vaccination as advised by learned reviewer.
  2. Author may consider adding details of SARS-CoV-2 or other viral infections specific changes in abundance of specific bacteria. 

Answer: Page 14: We believe this recommendation by the reviewer strengthens the significance of the article. We have added extra point (8.2. at Page 14) in the article and microbiome distribution has been articulated in Table 2 (Page 16) as well.

  1. It needs to be proofread for proper English grammar and small mistakes -a careful read through to correct these errors will significantly increase readability. Few Examples: Use of articles (a and the) are inconsistent throughout the manuscript  

Answer: I sincerely apologise for incorrect English and we have tried to correct it.

Term “The” ahead for many words, mostly in front of herd immunity has been removed throughout the manuscript.

  1. This process of natural vaccination is immensely pertinent to the current context of pandemic caused ….. context of “a” pandemic caused

Please find the changes:

  • Page 1 line 10- Abstract- This process of natural vaccination is immensely pertinent to the current context of a pandemic caused…..
  • Page 6; Fig 3 legend-……. during an epidemic or pandemic situations.
  • Page 12-during a pandemic or epidemic or endemic (……..
  • Page 14; 8.1. The fact that SARS-CoV-2 is certainly the cause of a pandemic but…
  • Page 17; 8.4. In a pandemic situation, the entire world is having a severe infection with SARS-CoV-2….
  • Page 19; Highlights: Point-3. ……..context of a pandemic scenario due to transmissible infection of SARS-CoV-2 virus.
  1. The conventional idea of herd immunity is established due to the efficient transmission of pathogens and developing the natural immunity in the populations…. Developing natural immunity in the populations.

Answer: Page 1- Abstract:  The word has been removed in front of word ‘natural’

  • This led to achieve the herd immunity in a particular cohort. … This led to achieving herd immunity in a particular cohort…

Answer: Page 1- Abstract:  “achieving” word has been replaced. Throughout the manuscript, “the” word has been removed ahead of ‘herd immunity.

  1. The microbiome play an essential ……. The microbiome “plays” an essential

Answer: Page 5- play word has been replaced with ‘plays’

Some sentence needs to be re-written as the same is showing similarity with text from earlier published article:

We have re-written as advised by the reviewer.

  1. “in the United States, around 20% of the city’s residents have been infected by the virus as of early May”,

Answer: Page 13 rewritten (Line 8-12)-  In the month of May, 2020, SARS-CoV-2  infection disseminated upto 20 % of total city dwellers of New York City, USA and this is the highest numbers of outbreak in a particular cohort in USA. Both the density of city dwellers in an area, and physical contacts among individuals are the critical factors to reach the threshold of herd immunity [125, 126]. Based on several experts, on average 60 % of the population should come in contact to the pathogen or its antigens for reaching herd immunity.

  1. “The herd immunity threshold may differ from place to place, depending on factors like”

Answer: Page 13 rewritten (Line 8-12) -  In the month of May, 2020, SARS-CoV-2  infection disseminated upto 20 % of total city dwellers of New York City, USA and this is the highest numbers of outbreak in a particular cohort in USA. Both the density of city dwellers in an area, and physical contacts among individuals are the critical factors to reach the threshold of herd immunity [125, 126]. Based on several experts, on average 60 % of the population should come in contact to the pathogen or its antigens for reaching herd immunity.

  • “could be higher. If there is a lot of variation in people’s likelihood of becoming infected when they are exposed, that could push the”

Answer: Page 13 rewritten (Line 8-12) In the month of May, 2020, SARS-CoV-2  infection disseminated upto 20 % of total city dwellers of New York City, USA and this is the highest numbers of outbreak in a particular cohort in USA. Both the density of city dwellers in an area, and physical contacts among individuals are the critical factors to reach the threshold of herd immunity [125, 126]. Based on several experts, on average 60 % of the population should come in contact to the pathogen or its antigens for reaching herd immunity.

  1. composition may predict predisposition to severe COVID-19 due to hyper-inflammatory response [103]. MRx-4DP0004, a strain of the bacterium Bifidobacterium breve,”

Answer: Page 14 rewritten Line 22-30-  The gut microbiome are the foremost factors to control the cytokine storm [130] inflammatory responses are controlled by specific microbiota composition, which may predict predisposition of COVID-19 [131] (Table 2). MRx-4DP0004, a strain of the bacterium Bifidobacterium breve under the phylum of Actinobacteria, originally developed for asthma [104], inhibited hyper-inflammatory response by diminishing the expression of angiotensin-converting enzyme 2 (ACE2) receptors for maintaining the potential anti-viral response [105]. The prevalent of SARS-CoV-2 receptor, ACE2 is highly expressed at microbiome enriched gut enterocytes and colonocytes (Wang et al., 2020; Xiao et al., 2020).

  1. “The specific memory T cell populations are preferentially re-expanded when re-exposed to the same antigen or one that is similar, thereby limiting the clonal expansion of new antigen-specific T cells (Fig. 2a) [106]. A similar mechanism has been proposed for B cell responses (Fig. 2b).”

Answer: Page 17 8.3. Line 5-13- rewritten-  Re-exposer of the same or similar type of antigen of latter pathogen stimulates and cross-reacts with memory T-cells which are specific for antigens of former pathogen or vaccination (Fig. 2a). Thereby, an immune repertoire memory T-cells have been preferentially re-expanded which refuted the clonal expansion of new antigen-specific T-cells. Consequently, the chances of generation of a new type of memory T-cells of latter pathogen will be thwarted unwittingly, known as ‘original antigenic sin’

  1. There is a lack of supporting data (i.e., bibliography) in some parts of the text. I would recommend the author reviewing the text and try to support some of the affirmations made. Reference needed for the following text
  2. Underprivileged children in the slam community commonly have excessive bacterial colonization at their proximal small intestine. It impairs the architecture of intestine with blunted villi, abnormal crypt to villus ratio.

Answer: Page 7:  adequate references included 

  1. Overburden of bacterial growth augmented the number of intraepithelial lymphocytes and a marked increase in lymphocytes which are the signature factors of allergic diseases.

Answer: Page 7:  adequate references included 

  • Therefore, the chances of herd immunity are quite futile in that population due to poor induction of specific innate and adaptive immune responses.

Answer: Page 7:  adequate references included 

  1. Live vaccines might then, instead of activating the innate immunity to enhance adaptive immune responses, be destroyed by an already highly activated leaky innate immune-mediated inflammatory responses.

Answer: Page 7:  adequate references have been included 

Reviewer 2 Report

  • please have the english language checked carefully as this is leading to some confusions
  • while the impact of the microbiome on immune responses in general is highlighted, for an review article on microbiome and vaccination (which is the most important way to induce herd immunity) discussion of that topic is underrepresented
  • please check if your statement is correct that the neonatal microbiome is composed of facultative anaerobes: Aagaard et al. have characterized a placental microbiome profile, composed of non pathogenic commensal microbiota from the Firmicutes, Tenericutes, Proteobacteria, Bacteroidetes, and Fusobacteria phyla which, interestingly, shares some similarities with the human oral microbiome (Aagaard K, Ma J, Antony KM, Ganu R, Petrosino J, Versalovic J. The placenta harbors a unique microbiome. Sci Transl Med (2014)
  • please refrain from broad assumptions for which no references are included or where it is not clear how you have arrived to those assumptions (e.g. under 3. you conclude that the "microbiome plays a pivotal role in mucosal immunity...." and link this to heard immunity without explaining how you arrive at this conclusion 

Author Response

Reply to second reviewer’s concerns:

We have worked on suggestions of erudite reviewer and answered all the issues.

  • please have the English language checked carefully as this is leading to some confusions

We, all authors appreciate the concern of deficiency of language, and we have tried to satisfy the reviewer by adequate corrections. We believe, first reviewer’s corrections enriched the quality of write-up further and it is fulfilling second reviewer’s concern.

Here is the point wise response to each question:

Major comments:

  • while the impact of the microbiome on immune responses in general is highlighted, for an review article on microbiome and vaccination (which is the most important way to induce herd immunity) discussion of that topic is underrepresented

Answer: Page 6-11: i. We have incorporated an extra section (4.1. to 4.3.) which discusses the potential role of microbiome on vaccination. We have also elaborated the role of microbiome on environmental enteropathy, probiotics-prebiotics and antibiotics on the efficiency of vaccination.

  1. We have included Table 1 which depicting the role of microbiota, environmental enteropathy, probiotics-prebiotics and antibiotics on different types of vaccination.

I hope this inclusion added the potential impact of the review article.

  • please check if your statement is correct that the neonatal microbiome is composed of facultative anaerobes: Aagaard et al. have characterized a placental microbiome profile, composed of non pathogenic commensal microbiota from the Firmicutes, Tenericutes, Proteobacteria, Bacteroidetes, and Fusobacteria phyla which, interestingly, shares some similarities with the human oral microbiome (Aagaard K, Ma J, Antony KM, Ganu R, Petrosino J, Versalovic J. The placenta harbors a unique microbiome. Sci Transl Med (2014).

Answer: Page 3: I do agree with this confusion. We have included pertinent references which makes the section essentialy relevant.

The total microbiome composition of infants are mostly occupied by all the phylums like Firmicutes, Tenericutes, Proteobacteria, Bacteroidetes, and Fusobacteria based on commensal microbiota harbors in placenta (Sood et al., 2006; Aagaard et al., 2014; Gritz et al., 2015). The variation of commensals is also related to mode of deliver of the infants. The intestinal flora of neonatal who born by vaginal delivery has resemblances with maternal vaginal flora. They are prevalent with Lactobacillus sp. and Prevotella sp. (Dominguez-Bello et al., 2010). In another report, vaginal delivery supports higher existence of facultative anaerobic organisms like Escherichia coliStaphylococcus sp.Streptococcus sp. and other Enterobacteriaceae (Phylum-Proteobacteria) for first few days of birth (Muleller et al., 2015; Pantoja-Feliciano et al., 2013). Nonetheless, caesarean babies are predominantly colonized with maternal skin microbiome like ClostridiumStaphylococcusPropionobacterium, and Corynebacterium (Biasucci et al., 2010; Madan et al., 2012). After born, milk feeding has significant impact of changing the microbiome composition of children. Breast milk provides improved prevalence of Bifidobacteria in intestinal track of infants. They are also occupied with higher abundance of Staphylococcus sp. Streptococcus sp., Lactobacillus sp. (phylum Firmicutes), Serratia sp., Ralstonia sp. (phylum of Proteobacteria), Corynebacterium (phylum Actinobacteria), etc in their gut (Hunt et al., 2011). On the contrary, there is richness of aerobic bacteria and lower frequencies of  Bifidobacteria in the gut of bottle milk-fed infants (Yoshioka et al., 1983; Balmer et al., 1989; Harmsen et al., 2000). Neonatal gut microbiota not only programs the metabolic function but also educates the naïve immune system, without that success of vaccination will be precluded.

  • please refrain from broad assumptions for which no references are included or where it is not clear how you have arrived to those assumptions (e.g. under 3. you conclude that the "microbiome plays a pivotal role in mucosal immunity...." and link this to heard immunity without explaining how you arrive at this conclusion 

Answer: Page 5 Line 13-17: We have added the references as advised by reviewer. We also modified the sentences.

In different oral vaccinations, microbiome plays a pivotal role in providing mucosal immunity [69]. Enhancement of mucosal immunity is the symphony of immunization process. Similarly, in OPV, intestinal microbiome has significant influence on mucosal immunity mediated adequate vaccine responses which is deficient in IPV where the route of immunization is parenteral [70].

Reviewer 3 Report

In their review manuscript, Shelly et. al. discuss microbiome, vaccination, and herd immunity in terms of what is known and unknown and provide a foil to the current COVID-19 pandemic. Overall, the topic of the manuscript is very interesting; however, the authors need to better shore up their hypotheses and tone down some of the more definitive and possibly inflammatory language to ensure scientific rigor.

  • The overall manuscript could use some editing for English grammar, language, syntax, and tense
  • Please ensure to define all acronyms upon first usage e.g. “SARS”
  • Please include a methods section that describes all literature searches, databases, Boolean terms, and inclusion and exclusion criteria for the literature reviewed in this manuscript.
  • Section 8.1 is difficult to understand as the authors frequently switch from talking about “threshold immunity” to “herd immunity” to “previous immunity” to having no immunity at all against SARS-CoV-2. In addition, the authors gloss over the point that there is still no strong evidence (aside from in vitro studies in monkeys) that previous SARS-CoV-2 infection generates lasting/protective antibodies. Indeed, now that reinfections are being observed, it may not even be advisable to be discussing herd immunity to SARS-CoV-2. Please consider revising extensively.
  • Section 8.3 is misleading and the authors utilize numbers to state that COVID-19 is almost “negligible” in India and hypothesize it is possibly due to microbiome or repeated vaccination. Please consider this section very carefully and avoid making inflammatory or incorrect statements, as this is a highly sensitive and unproven topic.
  • There is no direct, scientific proof of the statements the authors are making about the BCG vaccine and it is advantageousness to heterologous immunity to SARS-CoV-2. Pleae consider revising extensively so that statements reflect the current understanding of proven scientific literature and not just hypotheses.
  • While it is understandable that the global COVD-19 pandemic is a dynamic situation day-by-day, please try to ensure all numbers and statistic used in the article are as up to date as possible to ensure timeliness and accuracy.
  • The section on comparing the epitopes from the OPV vaccine and SARS-CoV-2 should be developed out into its own section and much more information needs to be added on the methodology and results.
  • The authors needs to make an overall better case for the connection between microbiome, vaccine, response, herd immunity, and SARS-CoV-2 – especially in the light of the fact there are several Phase III studies ongoing globally.

Author Response

Reply to third reviewer’s concerns:

We have worked on suggestions of raised reviewer and answered all the issues.

We, all authors appreciate all concerns and we have tried to satisfy the reviewer by adequate corrections. We believe, first and second reviewer’s corrections enriched the quality of write-up further and it is fulfilling third reviewer’s concern.

Here is the point wise response to each question:

Major comments:

  • The overall manuscript could use some editing for English grammar, language, syntax, and tense

We have improved the grammar, language. For syntax, we tried to break the sentences for appropriate  structuring.

  • Please ensure to define all acronyms upon first usage e.g. “SARS.

We have taken care of all the acronyms. Like, we have used ‘environmental enteropathy’ as EE from second appearance. 

  • Please include a methods section that describes all literature searches, databases, Boolean terms, and inclusion and exclusion criteria for the literature reviewed in this manuscript.

Page 18: Based on the overall suggestion by the reviewer, we have now made a separate section of the "In silico comparison of SARS-CoV-2 with OPV and BCG" where all protocol with different inclusion, exclusion criteria have been followed to extract data and run analysis is well described.

  • Section 8.1 is difficult to understand as the authors frequently switch from talking about “threshold immunity” to “herd immunity” to “previous immunity” to having no immunity at all against SARS-CoV-2. In addition, the authors gloss over the point that there is still no strong evidence (aside from in vitro studies in monkeys) that previous SARS-CoV-2 infection generates lasting/protective antibodies. Indeed, now that reinfections are being observed, it may not even be advisable to be discussing herd immunity to SARS-CoV-2. Please consider revising extensively.
    1. Here, our thinking differs with reviewer’s viewpoint. I apologise with variable opinion. We need to use word like “threshold/previous/poor” immunity to comprehend the degree of immunity which decides the efficiency of any vaccination or development of herd immunity.

We have changed the term ‘No immunity’ with poor immunity as your advice.

  1. However, in a pandemic, the concept of herd immunity is very much pertinent, especially when there is no medicine or vaccine available against SARS-CoV-2 till date. In this article, we have discussed about chances of success or failure of herd immunity generation and probable fate of the vaccine (heterologous immunity/original antigenic sin has been discussed) which is very much cardinal with the current pandemic.
  • According to reviewer 1, 8.1. paragraph has been rewritten.
  • Section 8.3 is misleading and the authors utilize numbers to state that COVID-19 is almost “negligible” in India and hypothesize it is possibly due to microbiome or repeated vaccination. Please consider this section very carefully and avoid making inflammatory or incorrect statements, as this is a highly sensitive and unproven topic.

Page 17 Section 8.4.: We appreciate the concern, raised by the reviewer, and we are partially agreed with it. Our study highlighted possible factors which might be associated with low infection and mortality rate, observed in SARS-CoV-2 pandemic.

To arrive at this conclusion, we have explored the total numbers of infection vs total numbers of tested population or pan India population according to  worldometer website (https://www.worldometers.info/coronavirus/).

According to data available on the website, India is at the second place in the globe, however we should also consider the total population of the country. The total deaths in India includes comorbidity, and there is always a high chance of mortality due to immunocompromised/autoimmune patients who are having infection with any pathogen. In spite of that, if India is having only 0.08 million of death including comorbidity, it raises the possibility that majority of population is somehow protective to the infection. We tried to connect the low infection rate observed in India with heterologous immunity.

However, there might be multiple reasons, explaining same. Thereby, we are not refuted that heterologous immunity is the only reason of resistance. We have hypothesized this theory logically on the basis of available literature, and data analysis. This will surely open the door for future studies.

  • There is no direct, scientific proof of the statements the authors are making about the BCG vaccine and it is advantageousness to heterologous immunity to SARS-CoV-2. Please consider revising extensively so that statements reflect the current understanding of proven scientific literature and not just hypotheses.

Page 18 Section 8.5.: We are agreed partially with the reviewer. We have added an extra point for better understanding.

  1. Several earlier reports have suggested heterologous immunity is critical factor providing resistance to many human viral infections. Thus, the theory of heterologous immunity should be taken into consideration while designing vaccines (Welsh et al., 2010; Zhou et al., 2012; Pusch et al., 2017). 
  2. Regarding the BCG's connection with heterologous immunity, there have been experimental studies well referenced in the manuscript like Arts et al., 2018 where BCG is providing protection for yellow fever which is a viral infection, through trained immunity. We have added one more reference (Jensen et al., 2015) where BCG vaccinated infants showed augmented production of cytokines upon heterologous challenge . Thus we think, BCG is providing heterologous immunity which might have implications in SARS-CoV-2 infection.

Thus, we did in-situ analysis to verify the hypothesis. We found protein sequence similarities between antigenic epitopes of SARS-CoV-2 and poliovirus, but not with BCG. We are doing further research to get the actual picture on it. We hope that further understanding of
the mechanisms might open up possibilities for disease prevention by
vaccination

  • While it is understandable that the global COVD-19 pandemic is a dynamic situation day-by-day, please try to ensure all numbers and statistic used in the article are as up to date as possible to ensure timeliness and accuracy.

Page 17 Section 8.4.: Yes, I am agreed with the learned reviewer. We have updated the number of COVID-19 patients up to September, 15, 2020. All the data has been taken from worldometer website (https://www.worldometers.info/coronavirus/) and further verified by Government of India official website (https://www.mohfw.gov.in/).

  • The section on comparing the epitopes from the OPV vaccine and SARS-CoV-2 should be developed out into its own section and much more information needs to be added on the methodology and results.

Page 18 Section 8.5: According to  reviewer's suggestion, we have made a separate section of OPV and SARS-CoV-2 epitope comparison (in silico comparison of SARS-CoV-2 , OPV and BCG) and have added the discussed organism IDs, the epitope location in the proteins, the protocol for extracting the epitopes, etc. in the revised manuscript.

  • The authors needs to make an overall better case for the connection between microbiome, vaccine, response, herd immunity, and SARS-CoV-2 – especially in the light of the fact there are several Phase III studies ongoing globally.

As advised by all three reviewers, we have done extensive editing with addition of two tables.

Page 3, 2.1; Page 6, 4.1. (added, advised by reviewer) with table 1; Page 10, 4.3.; Page 16- We are hopeful that all these additions and their further interpretations are strengthening the role of microbiome on vaccination and herd immunity.

Page 16, 8.2. with Table 2.- We also added the role of microbiome on COVID-19 .

Round 2

Reviewer 1 Report

N/A

Reviewer 3 Report

The authors have addressed my previous concerns to the best of their ability. Thank you.